# When Corona Infested Everything—A Qualitative Interview Study Exploring the Impact of COVID-19 Mitigation Measures on School Life from the Perspectives of English Secondary School Staff and Students

**DOI:** 10.3390/ijerph22060915

**Published:** 2025-06-10

**Authors:** Sarah Bell, Jane Williams, Jeremy Horwood, Sabi Redwood

**Affiliations:** 1The National Institute for Health and Care Research Applied Research Collaboration West (NIHR ARC West), University Hospitals Bristol, Bristol BS1 2NT, UK; j.horwood@bristol.ac.uk (J.H.); sabi.redwood@bristol.ac.uk (S.R.); 2Population Health Sciences, Bristol Medical School, University of Bristol, Bristol BS8 1UD, UK; 3School of Psychology, University of Bedfordshire, Luton LU1 3JU, UK; jane.williams@beds.ac.uk

**Keywords:** secondary schools, teachers, students, COVID-19, education, social behaviour, child welfare, mental health, psychological well-being, qualitative research

## Abstract

To reduce the risk of COVID-19 infection, transmission and illness during the pandemic, the Department for Education (DfE) issued guidance to schools. However, research on how the school community perceives the impact of the COVID-19 mitigation measures outlined remains limited. This qualitative study aims to explore the effects of school closures and in-school mitigation measures on daily school life, as well as their impact on mental health and wellbeing from the perspective of English secondary school staff and students. Participants were purposively sampled from English secondary schools serving diverse communities participating in the COVID-19 Mapping and Mitigation in Schools study (CoMMinS). Selection ensured representation of staff roles and student demographics. Semi-structured interviews were conducted remotely, and data analysed thematically. Interviews took place between January and August 2021 with participants from five secondary schools (20 staff and 25 students). Both staff and students reported significant disruption to school life, with four themes identified: (1) teaching and learning impact, (2) social impact, (3) safeguarding impact, and (4) and mental health and wellbeing impact. Findings highlight widespread negative effects across every aspect of school life, for both staff and students. This study suggests that COVID-19 mitigation measures in UK secondary schools led to a sense of loss and uncertainty as well as increased self-reported stress among both staff and students.

## 1. Introduction

### 1.1. COVID-19 School Closures and Mitigation Measures

The COVID-19 pandemic caused significant disruption to education [1]. To stop the spread of infection, English school closures were enforced by national government (March until September 2020 and January until March 2021), except for vulnerable children and those of key workers (defined by the government) [2]. For schools to operate safely and minimise COVID-19 transmission when open, Department for Education (DfE) guidance [3] required mitigation measures to be implemented, which included face coverings, a range of hygiene measures (sanitising and cleaning, respiratory and hand hygiene), COVID-19 test, trace and isolation procedures, increased ventilation, and numerous physical distancing measures to minimise contact within the school community.

### 1.2. Impact of COVID-19 Mitigation Measures on the Mental Health and Wellbeing of the School Community

The COVID-19 pandemic had a significant negative impact on young people’s mental health and wellbeing. There are consistent reports of increased levels of mental distress among students, including anxiety and depression, due to prolonged school closures and COVID-19 pandemic-related restrictions [4,5,6,7,8]. Young people themselves have expressed concerns about how the COVID-19 pandemic worsened their mental health and wellbeing [9]. Evidence suggests that those most negatively affected by COVID-19 and related restrictions experienced heightened emotional distress [10]. However, some studies also note that a minority of students—such as those who did not enjoy school or had pre-existing mental health difficulties—experienced improved wellbeing during lockdowns due to reduced social pressures and academic stress [10,11,12,13]. Qualitative UK studies highlight the broader impact on education and social interactions in addition to emotional struggles such as anxiety and depression [13,14,15,16,17,18]. The pressures of the COVID-19 pandemic also had a profound effect on school leaders [19,20,21] and teachers [22,23,24,25,26,27,28,29]. Concerns over high numbers of staff leaving the profession due to COVID-19 pandemic-related stress have been widely reported [30]. The increased workload from COVID-19 mitigation measures contributed to a decline in teacher mental health and wellbeing [22,23,24,25,26,27,28,29], with rising cases of anxiety [31] and depression [32] linked to job-related stress. Key stressors for school staff included organising and managing remote learning [23,25], the negative impact on teaching and student learning [24], navigating ongoing instability and change [1], addressing student wellbeing concerns [12,17], and coping with additional professional demands [7,8,9]. School leaders also faced financial constraints in implementing COVID-19 mitigation measures, as well as significant health and safety challenges when reopening schools [1]. Over time, evidence suggests that the growing demands of COVID-19 pandemic-related policy have contributed to a sustained decline in teacher mental health and wellbeing [33,34,35].

Despite an increasing body of research on the unintended consequences of COVID-19 mitigation measures in schools [36], there are few studies examining the impact of the DfE guidance and related measures on school life, and mental health and wellbeing from the perspective of the school community. The aim of this qualitative study was to explore how COVID-19 school closures and in-school mitigation measures affected school life, as well as their impact on the mental health and wellbeing of staff and students in English secondary schools. Staff and student perspectives on the effectiveness of implementing the mitigation measures, and whether the measures were effective at promoting safety, have been reported separately [37].

## 2. Methods

### 2.1. Study Design

A qualitative semi-structured interview study was conducted to explore the impact of COVID-19 mitigation measures on school life from the perspectives of secondary school staff and students.

### 2.2. Sampling Strategy

Staff and students were recruited from secondary schools in Southwest England participating in the ‘COVID-19 Mapping and Mitigation in Schools’ (CoMMinS) study [38]. Participants were purposively sampled to ensure a range of schools (size, deprivation level), staff roles (e.g., headteachers and teaching assistants) and student demographics (year group, eligibility for free school meals (FSM), ethnicity, and key worker/vulnerable group status) were included.

### 2.3. Data Collection

Semi-structured interviews were conducted remotely (using an online platform or telephone at a time suitable for the participant), using interview guides informed by the literature and DfE guidance (Appendix A for staff and student interview guide content). Staff received the study participant information via the school and contacted the researchers if interested in participating. Students received the study information via a recruitment video and website during a tutor, assembly, Personal Social and Health Education (PSHE) session, or they received the study website link via a school email. Potential participants were asked to register their interest through an online form and provide basic demographic information to inform purposive sampling.

### 2.4. Ethical Considerations

Written parental consent was sought for students under 16 alongside verbal assent from the student at the start of the interview (digitally recorded); verbal consent was sought for students over 16 years of age and for school staff. Students were given the option of participating in the interview with their parent/carer or peer from the same year and school. After the interview, all participants received a GBP 20 gift voucher to thank them for their participation. The study was approved by the Faculty of Health Sciences Research Ethics Committee at The University of Bristol (Ref. 112305).

### 2.5. Data Analysis

Interviews were digitally recorded, transcribed verbatim, checked for accuracy, and anonymised. The transcripts were analysed using a hybrid approach to thematic analysis [39]. Researchers (SB and JW) used line-by-line coding to construct initial draft coding frames, based on three transcripts, supported by QRS NVivo 12 qualitative data management software [40]. A combination of data-driven inductive and deductive coding, using pre-defined codes from the study objectives and interview guides (Appendix A), was used. To enhance analysis and maximise rigour, researchers (SB and JW) independently double-coded transcripts. These were compared and discrepancies were resolved by discussion. The refined coding frame was then applied to the remaining transcripts. Themes were developed, reviewed and refined, and a descriptive narrative of the data was created, to summarise the data under each theme. This process was completed separately for the school staff and student data.

## 3. Results

School staff and students were recruited from five secondary schools with diverse characteristics; four mainstream schools and one setting for students unable to attend mainstream (alternative provision). In total, 20 school staff (female: 12 (60%)) participated in interviews between January and July 2021 (including headteachers, senior leaders, class teachers and support staff). In total, 25 students participated in interviews between March and August 2021 (female: 14, (56%)). Two interviews were paired (with a nominated peer) and one comprised three peers; one interview was undertaken with support by a parent and another by an older sibling. Students ranged from Years 7 to 12 (aged 12 to 17) (Year 7 *n* = 7, Year 8 *n* = 2, Year 9 *n* = 3, Year 10 *n* = 6, Year 11 *n* = 1, Year 12 *n* = 6). Twenty eight percent were receiving FSM, 44% were white British, and 40% were in the key worker/vulnerable group attending school during lockdown closures. Table 1 shows the characteristics of participating schools (school size, range of year groups, percentage eligible for Free School Meals (FSM, proxy measure for deprivation), and Index of Multiple Deprivation (IMD quintile for each school) and number of participants interviewed in each school.

### 3.1. Overview of Findings

COVID-19 school closures and mitigation measures significantly disrupted all aspects of school life. Analysis of the data identified the following four themes:

Teaching and learning impact (Section 3.2): impact on teaching practices and processes, and student learning, exams, and the academic future of students;

Social impact (Section 3.3): impact on school culture and social opportunities, and social, emotional and behavioural needs of students;

Safeguarding impact (Section 3.4): impact on student safeguarding needs, and student welfare provision;

Mental health and wellbeing impact (Section 3.5): impact on staff mental health and wellbeing, and on student mental health and wellbeing.

### 3.2. Teaching and Learning Impact

Findings indicated that school closures and in-school mitigation measures impacted teaching and learning. Teaching practices and processes, and student learning, exams and their academic future were affected.

#### 3.2.1. Impact on Teaching Practices and Processes

School staff faced significant challenges in adapting their teaching methods and supporting student learning. During school closures, many struggled with using new IT systems and platforms to deliver and assess online lessons.

“It was a very quick sudden switch. A lot of us have been teaching for years and you come in each day and do your thing stood up in front of 30 faces each hour. Then all of a sudden, within a matter of days really, it was a switch to online learning. Just something that we’ve never had explicit training for that was very much learning on the job.” (P9, school 4, Assistant Headteacher)

However, despite the challenges staff faced in adapting their teaching methods, they also recognised and valued the development in teaching and learning that emerged over time due to advances in use of online technologies.

“The embracing of technology is probably the single biggest step forward for both adults and the children. We’re so, so, much better at it now, the live lessons we run, and the children understand how to use Teams.” (P8, school 4, Assistant Headteacher)

School staff frequently reported that in-school mitigation measures had a negative impact on teaching and learning on return to in-person education. Physical distancing requirements and face coverings made it difficult for staff to communicate effectively with students, support their learning, and assess their progress.

“Teachers were not to leave the 2-metre box at front of classroom, and they weren’t to circulate the room. So, a lot of pedagogy had to change. A lot of that live marking and that immediate assessment, some of that formative stuff, has not taken place in the same way, so that’s been a really difficult challenge.” (P6, school 2, Senior Leader)

Restrictions on student movement required staff to move between classrooms, which some found to negatively affect their teaching practice. Without a dedicated classroom, they described experiencing a sense of chaos and loss of control over their teaching environment. Staff also reported an increased teaching workload due to covering for colleagues absent due COVID-19 infection, isolation, or shielding. This added pressure left many feeling overworked and undervalued, with several describing their workload as unsustainable.

#### 3.2.2. Impact on Student Learning

All staff reported a learning deficit among students, commonly referred to as ‘lost learning’. This included delays in learning progress, loss of previously acquired knowledge, and missed curriculum content due school closures.

“We have these rolling schemes of work every year, we’ve just rewritten our whole curriculum, and we are having to pitch it a lot lower. That’s a good reflection of Year 7 behaviour and ability.” (P12, school 1, Class Teacher)

Many staff observed a lack of student engagement upon returning to school, often attributing this to students struggling to access the work. They expressed the greatest concern for students who had lower academic levels before the COVID-19 pandemic and those from disadvantaged backgrounds, noting that these groups were disproportionately affected.

“On-line learning definitely highlighted inequality. Students trying to do work on their phone, not having Wi-Fi, not having a laptop, not having any space to work. I’ve got some students caring for parents because parents are single and ill. It’s just an awful lot.” (P4, school 2, Senior Leader)

However, staff also highlighted some positive impacts of school closures on certain students, particularly those who typically struggle in a traditional school setting. Examples included students who were less confident, orthose with mental health challenges, autism, or behavioural difficulties, as they may have found remote learning environments less stressful and more accommodating of their needs.

Students echoed staff concerns about the negative impact of school closures on their learning, often attributing this to reduced academic support from teachers and missed curriculum content. Some also acknowledged that their own lack of motivation, decreased engagement, and reduced productivity during remote learning contributed to setbacks in their academic progress. Many students reported struggling with concentration, time management, and establishing a study routine while learning from home.

“One time my WiFi, it shut out like ten times during the online lessons and I was really preoccupied so I didn’t really have much motivation to do any of the work, I have no work from lockdown or anything, to be honest.” (Student 8, school 2, Year 7)

On return to school, students frequently expressed feelings of lost academic progress and competence, along with anxiety about catching up. Notably, no students reported any positive impact of school closures or in-school mitigation measures on their learning.

Both students and staff reported difficulties in coping with the changes and uncertainty over the exam processes and university application procedures. These disruptions caused significant stress for staff and heightened anxiety among students, many of whom worried about the impact on their academic future and job prospects.

“One of my Year 11 boys contacted me and he said, ‘Miss I’m just not sleeping and I’m finding it really hard to turn up for lessons’, and he’s my top student in that Year 11 group and I said, ‘Well you know are you going to bed at a reasonable time?’ and he said, ‘I’m just so anxious I wake in the night and I’m worried about my future and I’m worried about my grades.” (P12, school 1, Class Teacher)

Some students reported these difficulties led to apathy.

“It’s like everything got cancelled and everything got changed, so in terms of exam results and, you know, UCAS [university application process], there’s this kind of feel that it doesn’t really matter about choices because it will change.” (Student 24, school 5, Year 7)

### 3.3. Social Impact

Findings indicated that school closures and in-school mitigation measures had social impacts on staff and students. School culture and social interactions were disrupted, limiting opportunities for connection. Staff accounts also indicated that social, emotional and behavioural needs of students increased.

#### 3.3.1. Impact on School Culture and Social Opportunities

Both staff and students reported missing the culture of school life during school closures. On return to school, many felt that the overall ethos of schools had changed. Restrictions led to the loss of familiar rituals, routines, and norms, weakening their sense of community and belonging.

Social opportunities were also negatively affected, as many shared spaces and facilities were no longer accessible. Canteens and sports halls were repurposed for COVID-19 testing, while physical distancing measures limited the use of libraries, study areas, and social spaces. For staff, meeting rooms, shared offices, and staff rooms were also restricted, further limiting collaboration and connection.

“We can’t access the tennis courts or netball courts or anything that our school really has. We can’t even go to the library. Sport was a very big part of my school life and obviously it’s not anymore.” (Student 22, school 5, Year 12)

“We were all based in our offices having a good old catch up and good old grumble about everything, but I’ve hardly seen any of my colleagues at all in the last sort of three and a half weeks because you know we’ve been warned to stay out of the offices and not be around too many people so it’s quite isolating.” (P12, school 1, Class Teacher)

Students frequently reported the negative impact of missed extra-curricular activities, enrichment events, and other school-based experiences that typically support social development. The loss of these opportunities limited their ability to connect with peers, engage in new experiences, and develop important social skills.

“My teacher towards the end of our first year of sixth form, was like, ‘Okay guys, I can’t do this anymore. What we’re gonna do is next lesson, you bring in an object and you talk about it’. It was like a show and tell, it was something you do in Year 3. And yeah, I think that was quite awkward. But I think definitely people haven’t been talking as much, because we didn’t go out for, what, a good solid seven weeks.” (Student 21, school 5, Year 12)

“There’s no work experience and there are other things that we can’t do because of COVID which is probably having a negative impact on them.” (P15, school 1, Headteacher)

“Christmas performances, going away on residential trips and camps, all those kinds of things, which to be honest, are the big things for the kids that they always remember and enjoy, we haven’t been able to do.” (P1, school 3, Headteacher)

Students also expressed frustration over the negative impact of COVID-19 restrictions on their social lives beyond school. Many described feeling a lack of enjoyment, purpose, and motivation, as well as experiencing boredom due to the limitations on their daily activities.

“Like you get up and you know in your mind that you’re not gonna do anything that day—you’re just gonna sit at home and not do anything. It’s just like, ‘What am I doing with my life?’, ‘Why am I living this?’ Something that you just feel like, ‘I want to do something, but I don’t know what to do,’ if that makes sense. You don’t really have that much energy or motivation to do things. One day, I just would not feel like getting up or doing any exercise.” (Student 13, school 1, Year 7)

Students also lost out on important events that mark transitions and rites of passage in their school life as they had to be cancelled, highlighting the emotional impact of missing these important milestones.

“We had students leave us before summer without being able to say farewell to people. You know it’s loss of these experiences.” (P5, school 3, Senior Leader)

“It was a bit weird moving up (to secondary school) during the pandemic ‘cause we didn’t get open days and we didn’t get the headteacher talking to us face to face, she had to send us a video.” (Student 5, school 2, Year 7)

Staff experienced a significant loss of social interactions with colleagues which affected their ability to support one another and maintain a sense of teamwork during school closures and when returning under COVID-19 mitigation measures.

Some staff also reported a loss of professional identity due to the shift in their role as a teacher during school closures. The absence of in-person interactions with both colleagues and students left many feeling disconnected from their usual teaching environment. As a result, many staff members reported feelings of loneliness and isolation, further impacting their wellbeing and sense of belonging within the school community.

“In the September term that was quite miserable really, being stuck at the front and not being able to interact with people normally.” (P19, school 5, Senior Leader)

For some staff, the social impact of COVID-19 restrictions was particularly significant due to their personal living circumstances. Upon returning to school, many reported feeling unhappy as physical distancing measures limited their usual interactions with both colleagues and students. The cancellation of social and wellbeing activities and events, such as ‘coffee and cake’, further contributed to feelings of isolation, reducing opportunities for staff to connect and support one another.

“We normally have a barbeque in the summer where everyone brings their family and there’s a carol service and a commemoration day and there’s all these events that build up the culture and the environment of school that also really helps the staff. They used to meet for breakfast and this kind of stuff, that type of stuff is so important—so important and it will not be happening anywhere.” (P5, school 3, Senior Leader)

#### 3.3.2. Impact on Social, Emotional and Behavioural Needs of Students

All staff reported a rise in students’ social, emotional and behavioural needs. Many observed that students appeared unsettled, struggled to follow established school routines, and faced difficulties in managing peer relationships.

“The social skills they’re not as good as they were and we’re seeing more challenging behaviour from key students, and they show it in different ways. There’s lots more arguments between the kids in some classes, I guess that’s how they’re showing that they’re not okay.” (P7, school 3, Senior Leader)

Many staff highlighted that the increased social, emotional, and behavioural needs of students placed an additional burden on them, negatively affecting their own wellbeing and relationships with students.

“There’s a lot more negative interactions with children, just being unsettled and boisterous. They are really, really struggling to settle and I’m really upset because I’m not managing to meet the needs of the majority of the students because of the behaviour of the others and I’m so sick of it. It’s having not been in school for so long and being out of routine and who knows what they’re dealing with at home. Boredom, loneliness, other more serious concerns and you see it…you see it in them when they come back to school.” (P10, school 1, Class Teacher)

Staff also gave accounts of the challenges of physical distancing measures and face coverings, which made it more difficult to communicate effectively in the classroom. These barriers hindered their ability to manage behaviour and provide emotional support to students.

The disruption to key transition periods, particularly for Year 7 (moving to secondary school) and Year 12 (transitioning to sixth form), was especially concerning. Students in these year groups were not only below expected achievement levels but also lacked independence and essential social and emotional skills. As a result, both staff and students experienced greater difficulties, making school life more stressful and challenging.

“They’re not as mature as another Year 7 would be in previous years. They haven’t got emotional intelligence because they didn’t have the transition period.” (P11, school 4, Pastoral Support Teacher)

Echoing staff accounts, many students reported feeling unsettled on return to school. They struggled with developing a sense of independence during school closures and found it difficult to maintain friendships.

“The summer was ruined because I didn’t get to see any of my friends.” (Student 17, school 5, Year 7)

Students also described challenges in holding conversations with peers and ongoing relationship difficulties, often attributing these issues to restrictions on mixing with other students. Additionally, some experienced separation anxiety after spending an extended period at home with their families during school closures, making the transition back to school and taking part in extra-curricular activities even more challenging.

“Before COVID I could just go on a sleepover, go camping with my scouts or something. I wouldn’t be afraid to do it. But now, because I’ve spent so much time with my parents, I get nervous leaving them, I get nervous about going on sleepovers and doing stuff like that.” (Student 5, school 2, Year 7)

Despite the challenges, many students demonstrated resilience in adapting to post-lockdown school life. Staff reported that most students quickly re-adjusted to school routines, showing flexibility and the ability to cope with the disruptions caused by COVID-19. Their capacity to adapt highlighted their strength in navigating change despite the difficulties they faced.

“From what I’ve seen, students are remarkably adaptable and flexible and resilient and when they come back into the school building because that routine has been embedded, it’s remarkable how quickly it all seems normal again being back. And the same with staff, teachers are very routine orientated.” (P9, school 4, Assistant Headteacher)

### 3.4. Safeguarding Impact

Safeguarding in schools refers to the actions and policies put in place to protect students from abuse, harm, neglect, and exploitation. Findings indicated that school closures and in-school mitigation measures had a significant impact on safeguarding and student welfare. Staff reported an increase in safeguarding concerns, with a rise in student welfare needs during and after school closures, and a negative impact on safeguarding and welfare provision.

#### 3.4.1. Impact on Student Safeguarding Needs

During lockdowns, there were frequent reports of domestic violence and increased police involvement with families, highlighting the heightened risks faced by vulnerable students.

“There have been more domestic violence incidents at home; more parental conflict. There’s been conflicts between children and parents and abuse from children to parents, you know.” (P18, school 5, Deputy Headteacher)

Staff described being less able to identify risk and protect students from harm during school closures, resulting in an increase in need on return to school. Staff described feeling overwhelmed by the growing safeguarding demands, adding to their workload and emotional burden.

“I’ve heard really upsetting things while I’ve been on the phone to students. Really upsetting things that’s going on in a household, like domestic abuse. But also dealing with students in care that were significant risk. Obviously, my son was working at home, my husband was working at home, and I felt like I was bringing into my home all these safeguarding and wellbeing issues because I felt like I couldn’t escape them.” (P20, school 5, Senior Leader)

#### 3.4.2. Impact on Student Welfare Provision

Staff reported a significant loss of access to external services and professionals during the COVID-19 pandemic that were typically available previously to support students. Any support that remained available was often delayed and delivered online, which many felt was less effective and negatively impacted students in need. Frustration grew among staff due to the lack of welfare provision, as they took on increased responsibilities in identifying, monitoring, and supporting at-risk students.

“Social care still aren’t doing face to face, and it is still very much on the school to be sorting out these kind of issues. We only have a certain number of hours in the day where we can try and sort of sort their academic stuff out, and then to try and deal with everything that has been going on at home.” (P3, school 3, Senior Leader)

Many described feeling burdened by these additional duties, particularly during school closures when they had to step in to provide extra care. This included conducting home visits, delivering food packs, and making check-up phone calls to ensure student wellbeing and safety.

“It’s keeping children safe. We want to make sure these children (who normally have social workers) are okay and I think the emotional demand is massive, we don’t have the time to do what we feel is right, and that is really taking its toll.” (P1, school 3, Headteacher)

On return to school, staff reported feeling overwhelmed by the growing demand for pastoral care and mental health support. This was needed due to existing pre-COVID-19 pandemic backlog, an increase in new safeguarding concerns, and ongoing difficulties in accessing expert support services.

Staff also reported difficulties providing their usual pastoral care for students due to restrictions. Also due to pastoral staff being used to run COVID-19 testing and other health and safety duties.

### 3.5. Mental Health and Wellbeing Impact

COVID-19 school closures and in-school mitigation measures significantly disrupted school life, negatively affecting the mental health and wellbeing of both staff and students.

#### 3.5.1. Impact on Staff Mental Health and Wellbeing

Staff frequently reported feeling undervalued and overwhelmed by excessive workloads while lacking autonomy to make decisions. Accounts highlighted that the increased workload demands were unsustainable and raised concerns about staff retention. Teaching and supporting students became particularly challenging due to high learning needs, disengagement, and increased behavioural, social, and emotional difficulties. The lack of professional and external support further exacerbated these feelings. Additionally, concerns about COVID-19 exposure created a poor sense of safety, further impacting staff wellbeing (details reported in Bell et al., 2025 [37]). Many reported experiencing depression, anxiety, frustration, helplessness, and both mental and physical exhaustion.

“Staff have struggled. There has been an element of ‘I don’t feel like I can do this anymore’. So that feeling of loss of capacity.” (P8, school 4, Assistant Headteacher)

Most staff felt unsupported in managing their own mental health and wellbeing, both during school closures and after returning to in-person teaching. However, despite these challenges, staff in most schools remained committed to their roles, supporting one another and working together to navigate the difficulties.

“Everybody has a sense of—how can we do this and how can we achieve this for our students and ourselves. I don’t know, it’s a bit like you know in the war time they had all that sort of spirit, well we’ve kind of got that same spirit I suppose.” (P11, school 4, Pastoral Support Teacher)

#### 3.5.2. Impact on Student Mental Health and Wellbeing

Many students expressed resentment that the COVID-19 pandemic had become the central focus of school life, leaving them feeling burdened by its ongoing impact.

“For me it’s like every conversation you have it will somehow lead to Corona, and everything is infested with the subject Corona, every lesson we speak about Corona, every lesson.” (Student 7, school 2, Year 7)

There were frequent accounts of difficulties coping with loss, change and uncertainty with students describing a range of emotions including boredom, frustration, anger, detachment, disengagement, and loneliness. Many also felt let down, experiencing heightened anxiety, fear, stress, and depression. Some reported feeling so overwhelmed that it led to an increase in self-harm behaviours.

“It’s very hard to get out of negative thinking. A lot was triggered by COVID-19, but the effects are still there even without COVID being as strict. So, I do think it will impact a lot of people, like, further down the track kind of thing.” (Student 25, school 5, Year 12)

“I love school, I love my teachers and I love seeing my friends, I’m a very sociable person, I talked to the whole school, I talk to every person I see, so it was kind of hard for me. I was, ‘Oh, I’m so upset now’. It did make me depressed to be honest, I’m not gonna lie.” (Student 21, school 5, Year 12)

Additionally, students commonly experienced poor sleep, a loss of motivation, and a decline in their learning capacity, skills, and abilities. The lack of adequate welfare support—especially limited access to external professional services—further exacerbated their mental health challenges, leaving many without the help they needed.

## 4. Discussion

This study examined how COVID-19 school closures and in-school mitigation measures affected English secondary school staff and students. The findings revealed that disruptions to usual ways of working had a negative impact on school life, as well as the mental health and wellbeing of the school community. COVID-19 mitigation measures disrupted teaching and learning, impacted the school community socially, impacted safeguarding needs and welfare provision, and negatively impacted both staff and student mental health and wellbeing. Both staff and students experienced a strong sense of loss, uncertainty, and increased stress.

### 4.1. Implications for Staff

Participants’ accounts reinforced existing research highlighting the pressures faced by school staff [19,20,21,22,23,24,25,26,27,28,29,33,34,35]. Concerns about staff workload have long been recognised [41], and the additional burdens imposed by COVID-19 further intensified these challenges [42]. Reducing staff workload in future pandemics should be a key priority. This study also found that many staff felt unsupported in managing their own mental health and wellbeing during the COVID-19 pandemic. Addressing this issue is critical, especially given the well-documented link between staff wellbeing and student mental health [43].

### 4.2. Implications for Students

Previous research has consistently reported the negative impact of the COVID-19 pandemic on young people’s mental health and wellbeing [21,22,23,24,25,26,30,31,32,33,34,35], with most studies focusing on the effects of lockdown and school closures. More far-reaching consequences involving negative impacts on student behaviour, learning and pastoral care were predicted [44]. This study confirmed these showing that negative effects persisted beyond school closures during lockdowns and that in-school mitigation measures also had a significant adverse impact.

The latest national survey in England (2023) [45] reported an increase in mental health disorders in children and young people over time, suggesting that one in five now have a probable mental health disorder. Research also indicates that students themselves perceive their mental health and wellbeing as having deteriorated due to the COVID-19 pandemic [9]. Findings from this study provide further insight into how school closures and mitigation measures exacerbated issues. Feelings of stress, loss and uncertainty, and resentment of COVID-19’s disruption to school life have negatively affected students’ mental health and wellbeing. Given the established link between positive mental health and academic achievement [46], reducing disruption to school life in future pandemics and addressing students’ emotional responses are crucial for minimising long-term consequences on both wellbeing and academic outcomes.

Staff perspectives in this study also support concerns over the disproportionate impact of the COVID-19 pandemic on students from disadvantaged backgrounds, highlighting the need for targeted solutions to minimise these disparities in future pandemics. However, findings align with previous research suggesting that school closures had some positive effects for certain young people [10,11,12]. This raises the possibility that for some students, online learning could serve as a viable alternative to attending school under restrictive mitigation measures. Staff also reported that many students demonstrated resilience on return to school, a perspective that has been less frequently reported [47]. Further research into strategies for fostering student resilience and optimising remote learning is warranted.

### 4.3. Limitations and Strengths

This study was conducted between January and August 2021, capturing real-time perspectives of both staff and students during the COVID-19 pandemic. A key strength of this research is its ability to document the lived experiences of school closures and in-school mitigation measures as they unfolded. The triangulation of staff and student perspectives strengthens the validity of the findings by providing a more comprehensive understanding of the impact on school life and mental health and wellbeing.

Although the study sample was drawn from Southwest England, the diversity of school characteristics and COVID-19 rates comparable to other regions in England enhance the generalisability of the findings. Additionally, the implications for policy and practice may be relevant beyond the UK.

However, a longitudinal qualitative study would have provided deeper insight into both the immediate and long-term effects of the COVID-19 pandemic on school life, as well as the mental health and wellbeing of staff and students. Including parent perspectives may have further enriched the findings by offering additional viewpoints on the challenges faced by families during this period.

Despite these limitations, this is, to our knowledge, the first study to explore both staff and student experiences of school closures and COVID-19 mitigation measures in UK secondary schools. As such, it offers valuable insights to inform future pandemic preparedness and response strategies.

### 4.4. Further Research

Longitudinal studies would be valuable in assessing both the immediate and long-term effects of COVID-19 on school life, as well as on staff and student mental health and wellbeing. Further research could help identify effective interventions and strategies that have supported staff and students in coping with pandemic-related disruptions. These findings could inform evidence-based recommendations to mitigate negative impacts and enhance preparedness for future pandemics.

## 5. Conclusions

This study highlights the need for future pandemic responses to balance infection control with the wellbeing of school communities. The findings highlight the importance of minimising disruption to school life by maintaining routines, traditions, and key transition events. It is essential to reduce the negative impact of restrictions on social interaction and relationships, while strengthening mental health and wellbeing support for both staff and students. Continued access to specialist services and enhanced in-school capacity for safeguarding and pastoral care must be prioritised. The COVID-19 pandemic revealed that schools serve critical roles beyond education, providing essential support for welfare, mental health, and safeguarding. Future mitigation measures must be evidence-based to prevent avoidable harm to education and wellbeing, and to avoid deepening existing inequalities among students.

## Figures and Tables

**Table 1 ijerph-22-00915-t001:** Characteristics of participating schools.

Sample Characteristics	School 1	School 2	School 3	School 4	School 5
**N students in school**	<1000	<1000	<200	<1000	>1000
**Year groups**	Years 7–11	Years 7–11	KS2 * to Year 13	Years 7–11	Years 7–13
**N students interviewed**	5	8	<5	<5	8
**N staff interviewed**	5	<5	<5	<5	5
**% eligible for Free School Meals (FSM) ****	36.5%	12%	77%	34.9%	16.6%
**IMD 2019 quintile (within school type) *****	Most deprived (5)	Least deprived (1)	Most deprived (5)	Next most deprived (4)	Average deprived (3)

* KS2 indicates phase of learning (aged 7–12 years); ** FSM proxy measure for deprivation; *** IMD Index of Multiple Deprivation.

## Data Availability

The data underlying this article will be shared on reasonable request to the corresponding author.

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
