# Peer review of "When Corona Infested Everything—A Qualitative Interview Study Exploring the Impact of COVID-19 Mitigation Measures on School Life from the Perspectives of English Secondary School Staff and Students"

_ijerph, 2025, doi:10.3390/ijerph22060915_

Round 1
Reviewer 1 Report
Comments and Suggestions for Authors
Thanks for opportunity to review this manuscript as one of the IJERPH Academic/Associate Editors for children's environmental health and school health, safety, and wellness (especially secondary school teachers, administrators and students). This reviewer had substantive concerns to be addressed by authors, and this includes distinguishing what is new/novel about these 2020-2021 data in 2025, given so many scoping/systematic reviews cited + U.K. research cited from school-based studies, especially references #37-38 as listed. Please see attached PDF file; please focus on "sticky notes" pertaining to double-highlighted text and/or green pencil outlining text or parts of references list. The single-highlighted text just represents interesting points (by authors to/for me).

Author Response
Please see the attachment and details under 'reviewer 1'

Reviewer 2 Report
Comments and Suggestions for Authors
I must commend the efforts of the authors; however, some issues should be resolved before publishing the paper. They are:
(1) Since secondary schools were studied, I think it is better to use the word "students" instead of pupils.
(2) Looking at the abstract, the authors did not make it clear how the participants were selected. For instance, how many were selected from each school to arrive at the sample size employed in this study?
(3) For the students studied, were they in junior/senior classes, and how many were selected from each stream?
(4) How many staff were selected from each of the schools?
(5) The authors should justify their reasons for studying only females.
(6) The use of some slogans like guys, gonna, etc should not be allowed in this kind of study.
(7) There were no hypotheses stated in this study, and why?
Author Response
Please see attachment and responses to reviewer 2 section.

Reviewer 3 Report
Comments and Suggestions for Authors
The topic is acceptable, and methods are satisfactory by and large. Although, the presentation needs to be improved. My comments:
- Data gathering methods should be explained more in Abstract.
- Key-words should be selected from MeSH.
- Interview should be explained in detail.
- Any abbreviation in Tables like KS2 should be defined as a caption.
- Table-1 should be more clarified in the text.
- Results section is so boring and should be summarized.
- Name of the pandemic should be used consistently.
Author Response
Please see attachment and responses to reviewer 3 section

Round 2
Reviewer 1 Report
Comments and Suggestions for Authors
Thank you for your detailed point-by-point responses to each of 17 comments ("sticky note" in PDF of originally submitted manuscript) by me, reviewer #1. I also skimmed the revised manuscript with "track changes" and your supplement to see other minor edits you made in response to other reviewer(s), e.g., staff and students consistently specified, current supplement provided for paper (Word file).
Responses to comment #5 and #7 were the important or key clarifications and/or specific revisions related to other comments made by this reviewer, e.g., #2-3, #11, 14. Thank you for your explanations and subsequent minor edits in revised paper.
For comment #12, and comment #15: It is still not entirely addressed, the answer = yes! Please combine, into one good paragraph, major conclusions and recommendations for policy and practice based on this paper's/sub-study's qualitative data. [Note: The author's edit per reviewer #1 comment #13 is o.k.]
Author edits to other comments made by me/reviewer #1 are appropriate/good.
Author Response
Dear Reviewer 1, Sorry for the delayed revisions I have been unwell with a virus. The policy and practice implications and conclusions have now been combined into one good paragraph as suggested.
Just to highlight, on this system under 'title' it is still reading pupils although paper now using students instead (not sure if matters!) I have attached the updated paper below.
Best wishes Sarah
